# Lightweight Corn Leaf Detection and Counting Using Improved YOLOv8

**DOI:** 10.3390/s24165279

**Published:** 2024-08-15

**Authors:** Shaotong Ning, Feng Tan, Xue Chen, Xiaohui Li, Hang Shi, Jinkai Qiu

**Affiliations:** 1College of Information and Electrical Engineering, Heilongjiang Bayi Agricultural University, Daqing 163319, China; ningshaotong1999@163.com (S.N.);; 2College of Engineering, Heilongjiang Bayi Agricultural University, Daqing 163319, Chinajinkai2020_2023@163.com (J.Q.)

**Keywords:** maize, leaf counting, YOLOv8, lightweight, SatrNet

## Abstract

The number of maize leaves is an important indicator for assessing plant growth and regulating population structure. However, the traditional leaf counting method mainly relies on manual work, which is both time-consuming and straining, while the existing image processing methods have low accuracy and poor adaptability, making it difficult to meet the standards for practical application. To accurately detect the growth status of maize, an improved lightweight YOLOv8 maize leaf detection and counting method was proposed in this study. Firstly, the backbone of the YOLOv8 network is replaced using the StarNet network and the convolution and attention fusion module (CAFM) is introduced, which combines the local convolution and global attention mechanisms to enhance the ability of feature representation and fusion of information from different channels. Secondly, in the neck network part, the StarBlock module is used to improve the C2f module to capture more complex features while preserving the original feature information through jump connections to improve training stability and performance. Finally, a lightweight shared convolutional detection head (LSCD) is used to reduce repetitive computations and improve computational efficiency. The experimental results show that the precision, recall, and mAP50 of the improved model are 97.9%, 95.5%, and 97.5%, and the numbers of model parameters and model size are 1.8 M and 3.8 MB, which are reduced by 40.86% and 39.68% compared to YOLOv8. This study shows that the model improves the accuracy of maize leaf detection, assists breeders in making scientific decisions, provides a reference for the deployment and application of maize leaf number mobile end detection devices, and provides technical support for the high-quality assessment of maize growth.

## 1. Introduction

Maize is one of the important food crops and is a major contributor to the global food supply [1]. The leaf is an important organ for photosynthesis and nutrient production in maize. Therefore, the number of leaves is one of the key indicators for describing canopy development and growth. The irregular shape and distribution of leaves, as well as the influence of natural light, make the process of leaf segmentation and detection difficult. Inaccurate acquisition of plant phenotypic parameters may affect subsequent judgements of crop growth status and crop yield. This shows that maize leaves have a very important role in maize growth and development [2]. However, traditional maize leaf counting is mainly observed manually, which is inefficient and difficult, and is easily influenced by subjective judgement. Therefore, obtaining the number of maize leaves efficiently and accurately is important for analyzing maize growth. Plant phenotyping has become a hotspot, in which the total number of plant leaves is an essential morphological trait in phenotyping studies, especially for breeders; accurate leaf data help to select promising varieties and accelerate the breeding process. Due to the similarity of the appearance of the leaf blades while overlapping each other, the blades present different angles, etc., and these factors greatly increase the difficulty of the target blade identification, but also directly affect the accuracy and efficiency of the target blade feature extraction [3].

In recent years, with the continuous improvement in computer arithmetic power, deep learning techniques are widely used in the field of target detection. Applying deep learning to the processing of RGB images can provide a non-destructive and fast method for detecting corn leaves. Compared with the existing manual identification methods, the deep learning-based leaf number identification method has the advantages of low cost, high flexibility, high efficiency, etc. [4]. Through this method, agricultural managers understand the growth status of maize in a timely manner, and formulate scientific management measures, such as reasonable arrangements for weeding, fertilizer application, and pest control, to enhance maize yield and quality. The leaves of different plants are more different and can directly reflect the information of plant growth status. Li et al. [5] proposed a method to compute fruit normal vectors using edge computation and gradient direction distribution, and then proposed a fully convolutional single-stage instance segmentation network called LP3Net, based on feature prototypes, to target the occluded front view of litchi. The method achieves an average localization accuracy of 82%, which significantly improves the accuracy of lychee clustering harvest point localization. Liu et al. [6] proposed a soybean phenotype information sensing method based on improved YOLOv5, introducing MobileNetv2 to alleviate the backbone network, and introducing an attention mechanism and an improved loss function to improve the apparent robustness and generalization ability. The results show that the mAP of the improved YOLOv5 is 96.13%, and the correlation coefficient between the obtained data and the manually determined soybean phenotypic information is 0.98384, which has high accuracy and good consistency. Xu et al. [7] proposed a method for counting corn seedlings in the field combining semi-supervised learning, deep learning, and UAV digital images, constructing a segmented complete seedling model based on the semi-supervised learning framework Noisy Student’s SOLOv2, and detecting and counting corn leaves based on the Noisy Student’s YOLOv5x model, which effectively utilizes the semi-supervised learning framework to improve the performance of the model under data-scarce conditions. Zhang et al. [8] established a leaf recognition CNN model based on UAV images to estimate the rapeseed stand fraction from the number of recognized leaves. This study confirms that it is feasible to estimate the rapeseed stand fraction automatically, quickly, and accurately in the field, which provides a reference for phenotyping and cultivation management. Chen et al. [9] proposed a skeleton extraction and phenotypic parameter acquisition method based on the combination of target detection and key point detection models; this study can improve the accuracy of key point detection in single tiller rice plants and accurately acquire plant skeleton and phenotypic parameters, which can help rice breeding and improvement. Yu et al. [10] used UAVs to acquire corn tassel images from different periods, proposed a SEYOLOX-tiny model to accurately and robustly detect corn tassels, and embedded the attention mechanism to extend YOLOX to achieve key feature extraction to inhibit unfavorable factors. Barreto et al. [11] implemented automatic plant counting of sugar beets, maize, and strawberries based on a UAV-based camera system and a deep learning image analysis pipeline to automatically predict the number of crops per plot using a full convolutional network pipeline. Liu et al. [12] achieved high-throughput counting of maize seedlings based on UAV RGB images through corner point detection, linear regression and target detection. Zhong et al. [13] proposed an algorithm based on a Mask R-CNN deep learning network for segmentation and recognition of multi-targeted blades in complex backgrounds, and the results show that the recognition effect is good, the segmentation accuracy is 97.51%, and at the same time, the algorithm has a strong migration ability. Yi et al. [14] established a remote sensing detection method for fast and non-destructive estimation of the leaf age of maize seedlings, constructed an unmanned aerial vehicle (UAV) high-throughput phenotyping platform, and constructed a leaf age estimation model based on RGB and MS images, which provided strong technical support for observing leaf age in the field.

Currently, deep learning techniques are widely used in crop growth status monitoring. Aich et al. [15] used an inverse convolutional network for segmentation and a convolutional network for leaf counting to segment the leaves from the background and calculate the number of leaves, this method achieved better performance in segmenting the leaves from the whole background and calculating the number of leaves using simple data augmentation. Fan et al. [16] proposed a two-stream deep learning framework to segment and count different sizes of shaped plant leaves from images, achieving good performance on a public test set for automated analysis of plant phenotypes. Li et al. [17] developed a high-throughput method for counting the number of leaves by detecting leaf tips in RGB images; the Faster R-CNN model using a recurrent consistent generative adversarial network adaptive technique performed the best, and the self-supervised phenotyping method developed in this study offers great potential for solving a wide range of plant phenotyping problems. Guan et al. [18] proposed a network model based on the improved DBi-YOLOv8 and proposed the LTNS algorithm for counting maize leaves, the mAP and FPS of this model were 89.4% and 65.3%, which were 12% and 0.6% higher than the original model, respectively, and the method still has some limitations in dealing with leaf overlap and occlusion. Buzzy et al. [19] used Tiny-YOLOv3 for accurate real-time localization and leaf counting to create a simple robotics platform based on an Android phone and iRobot create2 to demonstrate the real-time functionality of the greenhouse network. Vishal et al. [20] calculated the total number of leaves by counting the number of leaf tips equal to the number of leaves and used the YOLO algorithm for detecting leaf tips as objects with an average accuracy of up to 82% and an IOU of about 0.53–0.60; the applicability of this method at different stages of growth still needs to be further validated. Gao et al.’s work [21] was based on improved YOLOv4 lightweight neural network for maize seedling number detection, using improved Ghostnet as the feature extraction network, introducing the attention mechanism and K-means clustering algorithm to improve the detection accuracy, and using depth separable convolution instead of ordinary convolution to make the network lightweight. Quan et al. [22] proposed an improved Fast R-CNN model to quickly and accurately detect maize seedlings at different growth stages under complex field operation environments, and the accuracy of this method was greater than 97.1% for both soil and weed detection in maize seedlings. Therefore, the rapid and accurate acquisition of maize leaf numbers is important to assist breeders in making scientific decisions and providing technical support for monitoring maize growth status.

Despite the success of the above studies in leaf identification and counting, there are still some problems. For example, there are fewer studies on maize leaf number detection and a lack of studies on lightweight maize leaf number detection with edge devices. Therefore, this study takes the maize growth process as the research object, and uses a deep learning network to achieve the recognition and counting of maize leaves with a low number of parameters, low latency, and high detection accuracy [23]. YOLOv5 and YOLOv8 are different versions of the one-stage target detection algorithms. YOLOv8, compared to YOLOv5, uses a more efficient feature extraction and fusion method employing deeply separable convolution, providing higher inference speed and lower latency for applications that require higher responsiveness. YOLOv8 introduces a new anchor mechanism that does not rely on manually preset anchor sizes, making the detection more flexible and efficient, and integrates a novel regularization technique in the training process to further improve the robustness and generalization of the model. In this study, based on the YOLOv8 model in the YOLO series, which has high accuracy, low participant count, and fast detection speed, we propose a lightweight method for recognizing and counting maize leaves, and develop a detection model that has high accuracy, fast detection speed, and is easy to deploy for mobile devices with limited computational power, which provides a feasible solution for maize growth detection for a more accurate assessment of the growth of maize.

## 2. Materials and Methods

### 2.1. Data Acquisition

The data for this study were collected at the Experimental Building of the College of Information and Electrical Engineering, Heilongjiang Bayi Agricultural University, Daqing City, Heilongjiang Province, China (125°9′50″ E, 46°34′58″ N). In this study, maize was grown and tested indoors from seedling to tassel stage. The image acquisition device uses a Xiaomi 14 mobile phone with a 50-megapixel camera with an imaging resolution of 3072 × 4096, which is capable of capturing detail-rich images to ensure the accuracy and reliability of the data acquisition, as well as ensuring consistency in the exposure and color of images taken at different angles to minimize the impact of light and background variations on the experimental results. Therefore, the multi-angle acquisition of maize at different moments from 8:00 to 18:00 from seedling to tasseling stage helps to capture the changes in various stages of the maize growth process, effectively avoids fluctuations in image quality due to changes in light, and provides high-quality image information for subsequent data analysis and research. Data were collected at the same time throughout the study period using an illuminance meter to determine light intensity, as well as a multi-angle collection path and a fixed collection angle to ensure that the angle and distance were the same for each collection. A total of 1000 images were taken as the experimental dataset, light and color corrections were performed at the data processing stage to reduce data fluctuations caused by environmental factors, images were annotated using labeling software, unqualified sample images were strictly screened, and samples with poor quality due to inconsistent lighting or changes in environmental factors were excluded to ensure a high-quality dataset. After the data labeling was completed, 80% of the samples were randomly selected from the samples to construct the training set and 20% for the validation set, respectively [24]. This division ratio ensures that the model has enough samples for training and helps the model learn the data features better to avoid overfitting, and 20% of the data can effectively evaluate the model’s performance on unseen data to ensure that the model has a good generalization ability. The dataset includes two categories: maize and maize leaves. During the data labeling and segmentation process, the labels were carefully proofread to ensure the accuracy of the data. Some of the datasets used in this study are shown in Figure 1.

### 2.2. Data Enhancement

This study employs data augmentation to perform various transformations and processes on the original images, generating more training data. This enhances the model’s ability to learn target features, improves the model’s generalization capability and robustness in different scenarios, and reduces the risk of overfitting. Setting the direction and intensity of the light source to simulate different solar lighting conditions increased the diversity of the data. Specifically, we set up three main lighting conditions: vertical lighting, angled lighting (30 degrees to 60 degrees), and horizontal lighting. Vertical lighting is used to simulate direct sunlight at midday, angled lighting to simulate morning and afternoon sunlight, and horizontal lighting to simulate low-angle light at sunrise and sunset. A large number of experiments were conducted to assess the impact of different data enhancement techniques on the model training results, comparing the performance metrics (e.g., precision, recall, etc.) of the models before and after the application of different enhancement methods, and finally random rotation, random brightness, saturation adjustments, random occlusion, noise, illumination variations, and blurring were used to enhance the training set [25]. These techniques help to generate diverse training samples and allow models to better adapt to different environments and changing conditions. During the enhancement process, each image is processed by randomly selecting one of the above-mentioned image enhancement methods. After processing, a new enhanced image is generated for each original image, resulting in 1000 enhanced images. After combining these images with the original images, the training set contains a total of 2000 images. Figure 2 shows the effect before and after data enhancement, from which it can be seen that the enhanced images have greater diversity in features, allowing the model to show better adaptability and robustness in practical applications.

### 2.3. The LCS-YOLOv8 Model Construction

YOLO (You Only Look Once) is a deep learning-based target detection algorithm proposed by Joseph Redmon et al. [26] in 2016. YOLO has been widely recognized and used in practical applications for its speed and efficiency. YOLOv8 is an improvement on the more popular YOLOv5 network, both in terms of detection accuracy and speed. YOLOv8 is an unanchored frame detection model that, instead of predicting the offset of the target from a known anchor frame, directly predicts the center of the target. Anchorless frame detection reduces the number of prediction frames as a way to speed up non-maximum suppression (NMS). The task-aligned assigner allocation strategy is mainly used in the YOLOv8 network; i.e., positive samples are selected based on the score-weighted results of classification and regression, and the loss computation covers both classification and regression branches. In this case, the classification branch was trained using binary cross entropy loss (BCE Loss), while the regression branch combined distribution focal loss (DFL) and complete intersection over union loss (CIOU Loss) to improve the accuracy of the model in predicting the bounding box. During model training, YOLOv8 turned off Mosiac enhancement for the last 10 epochs to improve model accuracy [27].

Aiming at the problems of a large number of model parameters, large weight files, and low detection accuracy, this study proposes a lightweight maize leaf count detection model based on improved YOLOv8. Optimizations and improvements were made in the following five main areas to achieve higher detection performance and lower computational cost [28]. Firstly, to solve the problem of oversized model weight files, a lightweight StarNet network [29] is introduced. The StarNet network achieves lightweight models by reducing the number of model parameters and the size of weight files. This not only reduces the storage and transmission costs, but also improves the deployment efficiency of models in resource-constrained environments. Secondly, in order to better deal with different varieties of maize plants, and considering that there are differences in the leaf shapes and sizes of different varieties that can lead to changes in the scale of maize images, the Channel Attention Fusion Module [30] (CAFM) was introduced into YOLOv8. The CAFM is able to dynamically adjust the weights of different channels so that the model better adapts to different scales and shapes of blades and improves the detection accuracy and robustness. Then, in order to better deal with the scale variation of target objects and complex background information, Large-Scale Separable Kernel Attention [31] (LSKA) is introduced to enhance the feature extraction capability and detection performance of the target detection model without significantly increasing the computational overheads, which splits the large size convolution into smaller convolution operations, which retains the advantages of large convolution and reduces the computational cost. In order to be able to capture more complex features while retaining the original feature information through jump connections, and to improve the training stability and performance of the network, the StarNet network is used to innovate the C2f module to obtain the C2f_Star module, which incorporates multi-level features so that the model performs well when dealing with targets at different scales, while maintaining efficient computational performance. Finally, in scenarios where computational resources are limited and efficient inference and deployment is required, a lightweight shared convolutional detection header (Detect_LSCD) is used to process the input feature maps to generate the bounding boxes and classification results for the targets. This approach not only reduces the consumption of computational resources, but also improves the speed of reasoning and is suitable for real-time monitoring tasks. In this study, the improved model is named LCS-YOLOv8 (Lightweight CAFM StarNet YOLOv8) network, and its model structure is shown in Figure 3.

#### 2.3.1. StarNet Lightweight Network

YOLOv8 uses CSPDarknet53 as the backbone network, a module with a deeper network structure, which, although it helps to improve the model’s ability to represent image features, makes model training and inference take longer due to its computational complexity [32]. Therefore, the backbone of YOLOv8 was replaced with the StarNet network, which utilizes an element-level multiplication method known as the “star operation”, which is similar to the kernel method in its ability to map inputs to a high-dimensional, non-linear feature space without expanding the network. The star operation can effectively fuse features from different subspaces, demonstrating high performance and efficiency. The implementation of the star operation in a single layer can be represented as an element-level multiplication of two linearly transformed features, by which a new feature space can be generated, significantly increasing the dimensionality without increasing the computational overhead. When applied to multilayers, it is able to recursively increase the implicit dimensionality and achieve almost infinite dimensional growth, which in turn improves the representation and performance of the network. The efficiency of the star operation is exploited to perform computations in the low-dimensional space while taking into account the high-dimensional features. This operation dramatically improves the network performance while keeping the network compact [27]. The StarNet network structure is shown in Figure 4.

#### 2.3.2. SPPF_LSKA Module

This study proposes an improved spatial pyramid pooling module SPPF_LSKA. This module combines the multi-scale feature extraction capability of SPPF and the long-range dependency capture capability of a Large Separable Kernel Attention module (LSKA) to achieve more efficient and accurate feature extraction. The LSKA decomposes the 2D convolutional kernels of a deep convolutional layer into cascaded horizontal and vertical 1D convolutional kernels, enabling the direct use of the deep convolutional layer of a large kernel in the attention module without the need for an additional module. The LSKA module enhances remote dependency of the input image without leading to a large amount of computation and memory usage. The original SPPF extracts features through three maximum pooling layers and feeds the output after joining these features into the convolutional layer for processing, whereas the improved SPPF_LSKA extracts features through three maximum pooling layers and joins these features and outputs them into the LSKA module for processing and then feeds the output into the convolutional layer for processing. The LSKA module captures a larger range of spatial features through large kernel convolution, enabling the network to effectively focus on key regions and enhance feature extraction capabilities; the LSKA module is applied to the connected feature map output from the maximal pooling layer to better fuse multiscale features and enhance the ability to express important features [29]. The structure of the SPPF_LSKA module is shown in Figure 5.

#### 2.3.3. CAFM Attention Module

Various attention mechanisms have emerged to improve the accuracy of target detection tasks. Existing models may suffer from insufficient feature expressiveness when dealing with multi-scale targets and complex backgrounds. Convolutional operations are limited by local properties and senses and are deficient in modeling global features, so a convolution and attention fusion module is proposed. The CAFM module enhances feature representation by combining convolutional operations and attention mechanisms and improves overall performance by weighting features so that the model focuses on important information. In this study, the CAFM attention module is introduced into the head of the YOLOv8 model structure to enhance feature representation by capturing long-range dependencies and local feature extraction. The proposed CAFM consists of local and global branches. The channel dimensions are first adjusted using 1 × 1 convolution and the output tensor of each group is connected along the channel dimensions to generate a new output tensor. Then, features are extracted using 3 × 3 × 3 convolution. The local branches are shown in Equation (1):(1)$$F_{conv}=W_{3\times 3\times 3}(CS(W_{1\times 1}(Y)))$$
where *F_conv_* is the output of the local branch, *W*_1__×1_ denotes 1 × 1 convolution, *W*_3__×3__×3_ denotes 3 × 3 × 3 convolution, *CS* represents channel shuffle operation, and *Y* is the input feature. In the global branch, we first generate query(Q), key(K) and value (V) via 1 × 1 convolution and 3 × 3 depth-wise convolution, yielding three tensors with the shape of $\overset{\wedge }{H}\times \overset{\wedge }{W}\times \overset{\wedge }{C}$. The output *F_att_* of the global branch is defined as shown in Equation (2):(2)$$F_{att}=W_{1\times 1}Attention\left(\overset{\wedge }{Q}\right.,\overset{\wedge }{K},\left.\overset{\wedge }{V}\right)+Y$$
(3)$$Attention\left(\overset{\wedge }{Q}\right.,\overset{\wedge }{K},\left.\overset{\wedge }{V}\right)=\overset{\wedge }{V}\;Softmax\left(\overset{\wedge }{K}\right.\left.\overset{\wedge }{Q}/\alpha \right)$$
where *α* is a learnable scaling parameter to control the magnitude of matrix multiplication of $\overset{\wedge }{K}$ and $\overset{\wedge }{Q}$ before applying the softmax function. Finally, the output calculated by the CAFM module is shown in Equation (4):
(4)$$F_{out}=F_{att}+F_{conv}$$
The CAFM [28] network structure is shown in Figure 6.

#### 2.3.4. C2f_Star Feature Extraction Module

The C2f module suffers from insufficient feature extraction and large model parameters in maize leaf detection. This study proposes a new C2f_Star module based on CVPR2024-StarNet for C2f innovation. The C2f_Star module combines the Star_Block module on the basis of retaining the structure of the original C2f module, introducing a more efficient feature extraction and fusion method, which improves the accuracy and speed of detection. In the Star_Block module, the convolution operation is implemented by depth-separable convolution, which decomposes the standard convolution into depth convolution and point-by-point convolution, reducing the number of computational individual parameters. The jump connection and ReLU6 activation function alleviate the problem of gradient vanishing and explosion and promote the gradient flow for better model convergence. Convolutional kernels of different scales are introduced to the feature extraction unit, which can capture more levels of information; the feature fusion unit adopts an adaptive weight allocation method based on the attention mechanism, which effectively combines features of different scales [27]. The C2f_Star module enables the network to significantly improve the accuracy and robustness of feature extraction while maintaining computational efficiency by introducing the Star_Block module, adapting to different task requirements. The C2_Star network structure is shown in Figure 7.

#### 2.3.5. Lightweight Shared Convolutional Detection Header Detect_LSCD

The raw detection header of YOLOv8 contains multiple detection layers, each of which has its own independent convolution operation for processing the input feature maps and generating the final detection results, which results in a large number of parameters and computation due to the independent convolution operation of each detection layer. Based on the above problems, a lightweight shared convolutional detection head (LSCD) is proposed, which effectively reduces the number of parameters and computational overheads by sharing convolutional layers. Detect_LSCD uses two shared convolutional layers to process the feature maps of all detection layers [33]. This sharing mechanism not only reduces the number of parameters in the model, but also improves the inference speed. Meanwhile, the multi-layer detection mechanism of YOLOv8 is retained, and each layer processes the feature map through specific convolutional and scaling layers to ensure the high accuracy of the detection. The structure of the Detect_LSCD network is shown in Figure 8.

### 2.4. Model Evaluation Indicators

In order to assess the accuracy and efficiency of the improved YOLOv8 algorithm in maize leaf detection and counting, this study evaluates and analyzes the performance of the model from different perspectives using the following metrics.

Accuracy and recall are the most commonly used evaluation metrics in target detection. Accuracy is defined as the ratio of the number of correct samples in a test result to the total number of samples in all test results, and recall is defined as the ratio of the number of correct samples in a test result to the total number of real samples. Typically, accuracy and recall are mutually constraining metrics, which means that a detector may increase accuracy and decrease recall, and vice versa. *Precision* (P) and *Recall* (R) are defined as in Equations (5) and (6):(5)$$Precision=\frac{TP}{TP+FP}$$
(6)$$Recall=\frac{TP}{TP+FN}$$
where true positive (*TP*) and false positive (*FP*) denote the number of samples correctly predicted to be in the positive category and the number of samples incorrectly predicted to be in the positive category, respectively. False negative (*FN*) denotes the number of samples that were incorrectly predicted to be in the negative category.

The *F*1 score is a combination of accuracy and recall, which is used to evaluate the detection results of the model. The *F*1 score is obtained by calculating the sum mean of accuracy and recall. The higher the *F*1 score, the better the detection ability of the model. The *F*1 score is defined as shown in Equation (7):(7)$$F1=\frac{2\times Precison\times Recall}{Precision+Recall}$$

In the case where there is more than one detection category, the mean accuracy (*mAP*) is a more comprehensive representation of the model’s performance, and by calculating and averaging the *mAP* for each category, a more comprehensive evaluation metric can be provided. The process of calculating *mAP* consists of sorting each category according to the confidence of the prediction frame, calculating the average precision of each category according to the accuracy–recall curve, and then averaging the average precision of all the categories to obtain the final *mAP* value. Higher *mAP* values indicate the better detection ability of the model. The *mAP* is defined as shown in Equation (8):(8)$$mAP=\frac{1}{N}\sum_{i=1}^{n}AP_{i}$$
where *N* is the number of categories and *AP_i_* is the *AP* of the ith category.

Floating-point operations per second (FLOPs) represent the number of floating-point operations performed per second, which can characterize the complexity of the model. When the network structure is more complex, the neural network model’s FLOPs value is higher, which can reflect the scale of computational resources required by the model to some extent.

Parameters reflect the number of parameters in the leaf number detection model. Parameters can affect the training time, storage size, and detection performance of the leaf number detection model and can be used to measure the complexity of the model.

## 3. Results and Analysis

### 3.1. Experimental Environment

This study is based on the open-source machine learning framework Pytorch 1.11.0. The server experimental environment is Ubuntu 20.04, the processor is 16 vCPU Intel(R) Xeon(R) Gold 6430, the memory is 120 GB, the graphics card is RTX4090 with 24 GB video memory, and the GPU acceleration environment is Cuda 11.3. The Python version is 3.8. The training parameter settings are shown in Table 1.

### 3.2. Performance Comparison of Similar Models

In this study, a series of new improvements are made to the original YOLOv8 model to improve the accuracy and efficiency of target detection. The performance of the new network is evaluated by experimentally comparing it with other mainstream models. The comparison metrics include precision, recall, mean average precision (mAP), number of floating-point calculations, number of parameters, and model size. The comparison results are shown in Table 2.

As can be seen from the experimental results in Table 2, the Faster R-CNN and SSD models have larger floating-point computation and parametric quantities and generate larger model weight files. Therefore, the Faster R-CNN and SSD algorithms are less suitable for the lightweight real-time detection requirements of this dataset. The Faster R-CNN model has 941.0 G of FLOPs, 28.3 M of parameters, a model size of 108.2 M, and an FPS of only 85.6, while the SSD model has 60.9 G of FLOPs, 23.8 M of parameters, a model size of 91.1 M, and an FPS of 103.4. These properties make these two models efficient in resource-constrained environments. The YOLOv9 model was lower than YOLOv5s, YOLOv8n, and LCS-YOLOv8 in terms of precision, recall, F1, and mAP50, and the FPS was reduced by 43.4 compared to LCS-YOLOv8. The accuracy of YOLOv5s is slightly higher than that of YOLOv8n, but the number of parameters and the model size of YOLOv5s are about twice as large as that of YOLOv8n, and the FPS is reduced by 67.1 compared to LCS-YOLOv8. The improved algorithm LCS-YOLOv8 improves precision and recall by 0.1% and 0.3% and FPS by 6.6 compared to the original YOLOv8n, and the number of parameters of the improved model and the model weight file are smaller, which makes it more suitable for use in resource-constrained environments. The improved algorithm LCS-YOLOv8 was compared to YOLOv5s; although the precision and recall are slightly reduced, the number of parameters and the model size of YOLOv5s are about three times those of LCS-YOLOv8, and the FPS is reduced by 67.1 compared to LCS-YOLOv8. As can be seen from the above analysis, the LCS-YOLOv8 algorithm proposed in this study demonstrates its superiority across a number of metrics, substantially reducing the model complexity and computational requirements while maintaining high detection performance. This makes LCS-YOLOv8 an efficient and reliable choice for resource-constrained environments, capable of achieving faster processing speeds while maintaining accuracy.

### 3.3. Ablation Experiment

The original YOLOv8 model FLOPs and model size are large, so an improved YOLOv8 lightweight neural network is proposed. The specific improvement method is as follows: the StarNet network is used as the backbone feature extraction network, and different subspace features are fused by star operation in order to reduce the number of network parameters. Using an improved spatial pyramid pooling module, the LSKA module captures spatial features through large kernel convolution, focusing on key regions to enhance extraction. The CAFM attention module is used to focus on important information through weighted features to improve the overall performance. Using deep convolution can separate and share the convolutional detection head to reduce the number of network parameters. In order to verify the effectiveness of each improvement module in the algorithm of this study, the original model YOLOv8 is used as the baseline model, and recall, F1 score, mAP50, number of floating-point calculations, number of parameters, and model dimensions are used as the evaluation indexes, and ablation experiments are carried out through different combinations of multiple improvement modules, and the results are shown in Table 3.

Based on the analysis of the experimental results in Table 3, it can be seen that YOLOv8 performs well in terms of recall, F1 score and mAP50, but its FLOPs and parameter sizes are large and unsuitable for resource-constrained environments. The accuracy of the improved StarNet modular network remains unchanged from YOLOv8, and the FLOPs, number of parameters and model size are reduced by 1.6 G, 0.8 M, and 1.6 MB, and the FPS is improved by 1.5. The experimental results show that StarNet’s lightweight design may lead to inadequate feature extraction affecting the recall rate, and that the use of deeply separable convolutional modules for the backbone network greatly reduces the number of parameters and floating-point computations that the model generates during ordinary convolution.

In the improved StarNet module and LSCD module networks, the accuracy is reduced by 0.5%, recall is improved by 0.5%, FLOPs are reduced by 2.4 G, the number of covariates is reduced by 0.9 M, the model size is reduced by 3 MB, and the FPS is reduced by 3.2, when compared to the YOLOv8 network. Compared to the improved StarNet network, precision is reduced by 0.5%, recall is increased by 0.7%, FLOPs are reduced by 0.8 G, the number of covariates is reduced by 0.1 M, and model size is reduced by 0.4 MB. The experimental results suggest that due to the shared convolutional nature of the LSCD module, the individual detection layer features may not have been adequately separated leading to a decrease in precision, but the lightweight design of the LSCD may have made the detector more sensitive to the target improving the recall. Adding the LSKA module on top of this increases precision by 0.6%, decreases recall by 0.5%, increases FLOPs by 0.5 G, decreases the number of parameters by 0.2 M, decreases model size by 0.4 MB, and increases FPS by 3.8. The experimental results show that the LSKA large convolutional energy can cover a larger range of input features and identify more details and contextual information to improve the accuracy, the LSKA module can maintain a good recognition of potential targets, and the recall ability of the model is not affected by the structural changes.

Compared to the original YOLOV8, the improved StarNet module and LSKA module add a 0.2% increase in C2f-Star module accuracy, a 2.7 G reduction in FLOPs, a 0.7 M reduction in the number of parameters, a 1.5 MB reduction in the model size, and a 2.7 increase in FPS. Compared to the improved StarNet and LSCD modules, the accuracy is increased by 0.7%, FLOPs are reduced by 0.3 G, the number of parameters is increased by 0.2 M, the model size is increased by 0.5 MB, and the FPS is increased by 5.9. The experimental results show that the C2f-Star module enhances the ability of features to work together at different scales, conveys and fuses feature information to improve model accuracy, significantly reduces the number of model parameters due to the use of a large number of depth-separable convolutions, and reduces the computational complexity of inference to improve inference speed.

The improved CAFM module has a precision reduction of 0.1%, a recall increase of 0.2%, a reduction of 3.1 G in FLOPs, a reduction of 1 M in the number of parameters, a reduction of 2.2 MB in the model size, and an increase of 3.9 in the FPS compared to the original YOLOv8. Compared to the improved StarNet module, recall increased by 0.4%, FLOPs decreased by 1.5 G, parameter count and model size decreased by 0.2 M and 0.6 MB, and FPS increased by 2.4. The experimental results show that the CAFM module uses 3D convolution to obtain feature information more comprehensively from different scales and dimensions to increase recall, and the deep and grouped convolution in CAFM significantly reduces the computational complexity, allowing the model to be computed faster in the inference process.

The LCS-YOLOv8 model shows a 0.6% improvement in accuracy, a 3.2 G reduction in FLOPs, a 1.2 M reduction in the number of parameters, 2.5 MB reduction in the model size, and a 6.6 improvement in FPS compared to the original YOLOv8 model. Compared with the improved CAFM module, the precision is improved by 0.2%, recall is improved by 0.1%, FLOPs are reduced by 0.1 G, the number of parameters is reduced by 0.2 M, the model size is reduced by 0.3 MB, and FPS is improved by 2.7. The experimental results show that the LCS-YOLOv8 model is able to better capture and represent complex features in images, capture targets at different scales through multi-scale feature fusion, achieve efficient feature processing and fusion using a smaller number of parameters and computations, and reduce the number of parameters and size of the model. As a result, the LCS-YOLOv8 model occupies less memory and consumes less energy on mobile devices, making it more suitable for edge computing and cloud computing deployments.

### 3.4. Model Generalization Test Experiment

The visualization provides a more intuitive view of the effectiveness of the algorithm in detecting corn leaves. It allows for observing information such as the position, size, and category of the corn leaves detected by the algorithm, which can be used to evaluate the accuracy and efficiency of the algorithm, as well as to debug and optimize the algorithm. According to the ablation test results, by selecting one of the better model algorithms for comparative analysis and validation, YOLOv5s, YOLOv8, and the improved LCS-YOLOv8 algorithm of this study were selected to visualize and compare the detection results in the maize growth.

#### 3.4.1. Comparison of Monitoring Effects of Different Models in Different Light Environments

To validate the generalization of the proposed model for maize leaf detection under different light conditions. Figure 9 shows some of the detection results, group A shows the maize taken under normal conditions, group B shows the maize taken under a simulated cloudy day, and group C shows the maize taken under a simulated sunny day, and one maize image is selected from each of the three scenarios to be shown. Based on the comparison of the test results of A, B, and C, it can be seen that the three models are more capable of detecting under normal weather conditions, and YOLOv8 has leakage detection for leaves with different characteristics under normal conditions. In cloudy and sunny conditions, YOLOv8 and YOLOv5s’s detection abilities are slightly reduced compared to normal conditions, but they still maintain a high level of accuracy. It can be seen that different light intensities have a certain impact on the detection results. The improved LCS-YOLOv8 algorithm performs stably and has high recognition accuracy in three weather conditions, which proves that the algorithm in this study can improve the problem of inaccurate target feature recognition in different weather conditions.

#### 3.4.2. Comparison of the Detection Effect of Different Models in Different Growth Periods

To verify the generalization of the proposed model for maize leaf detection at different growth periods. In this study, the leaf images of maize at the 4-leaf stage, 7-leaf stage and 9-leaf stage were randomly selected from the test set and input into YOLOv5s, YOLOv8n, and LCS-YOLOv8 for experiments, respectively, to observe the confidence situation, and the results of detection are shown in Figure 10. Among them, the confidence level of LCS-YOLOv8 for the leaf detection frames at 4-leaf stage and 7-leaf stage was around 0.9, which was improved compared to YOLOv8 and YOLOv5s; the confidence level of YOLOv5s was lower and there was a leakage of detection at the 7-leaf stage compared to LCS-YOLOv8 and YOLOv8 leaf detection frames. In the detection of leaf images at the 9-leaf stage, the LCS-YOLOv8 not only maintains high accuracy, but also has no leakage occurrence, and the YOLOv8 and YOLOv5s miss one leaf at the 9-leaf stage. Higher confidence in the detection frame indicates that the model has better captured the detailed features of the target during the learning process, and a higher likelihood that the target exists. Due to the continuous growth of maize leaves, the characteristics of cotyledons change considerably. As a result, YOLOv8 and YOLOv5s missed the yellowing cotyledons during detection. The LCS-YOLOv8 model embedded in the CAFM attention module proposed in this study can highlight important leaf information and improve the expression of different features, thus better capturing maize leaves. The experiments illustrate that the improved LCS-YOLOv8 has a better detection effect compared to the original YOLOv8 model, and this conclusion provides technical support for subsequent maize leaf number detection.

### 3.5. Feature Visualization of Different Networks

To further explore the feature extraction capabilities of different networks, the feature maps are visualized. The visualization allows an intuitive look at the features extracted by the model during the detection process and helps to understand its detail capturing capabilities. The final feature extraction results for YOLOv5s, YOLOv8, and LCS-YOLOv8 networks are shown in Figure 11.

The comparison shows that the feature map of the LCS-YOLOv8 model can clearly present the characteristics of the leaves. The edges, textures, and details of the blades are better preserved and highlighted in the feature maps of LCS-YOLOv8, while the blade features are relatively blurred in the feature maps of YOLOv5s and YOLOv8. This difference suggests that the LCS-YOLOv8 model has a greater ability to capture more detailed leaf features in terms of feature extraction. In summary, the LCS-YOLOv8 model is able to learn more leaf features and shows an excellent feature extraction ability and detection effect compared with YOLOv5s and YOLOv8. LCS-YOLOv8 is more suitable for maize leaf detection and provides reliable technical support for maize leaf detection.

## 4. Discussion

Target detection has a wide range of applications in agriculture, such as pest and disease detection, crop identification, and plant leaf counting. The number of leaves is an important piece of phenotypic information during maize growth. In this study, the improved YOLOv8 lightweight network model can accurately obtain the number of maize leaves to understand the growth rate and health status of maize, which can help to assess crop yield and specify reasonable field management measures.

Intact single maize plants are a prerequisite for accurate calculation of the number of maize leaves. In this study, the lightweight LCS-YOLOv8 model was chosen to compare with traditional one- and two-stage target detection algorithms, respectively. As can be seen from Table 2, LCS-YOLOv8 not only approaches or even exceeds other mainstream models in terms of precision and recall, but also excels in terms of computational complexity, number of parameters, and model size. LCS-YOLOv8 is able to significantly reduce computation and storage costs while ensuring detection accuracy, making it suitable for resource-constrained application scenarios. In contrast, while the two-stage target detection algorithm performs well on some metrics, its high computational complexity and large model size do not provide a competitive advantage in resource-limited environments. The experiments showed a 5.13% increase in precision and a significant increase in recall and F1 values compared to Wang et al. [34] when using the LCS-YOLOv8 model to detect and count maize leaves. In addition, the lightweight shared convolutional detection head processes the feature maps of all detection layers by sharing two convolutions, which significantly improves precision, recall, and increases FPS by 53.3 compared to Ye et al.’s [35] use of lightweight YOLOv8. Therefore, for the application of corn leaf identification and counting tasks in lightweight scenarios, the improved model performs well in resource-constrained environments, but there is still room for improvement in detection accuracy under extreme conditions (e.g., sunny and cloudy days). Future work will further improve the robustness and adaptability of the model.

The method proposed in this study was evaluated only on a single maize dataset, and after training it achieved a mAP of 97.5 in the test set containing 127 images, which was in addition to a good performance. However, the effectiveness and performance for other periods (e.g., irrigations) or population maize detection tasks need to be further explored. In the future, it is planned to extend the applicability of the model by adding large field maize data to the training set and to validate its effectiveness in real production through maize detection in pots in greenhouses. Due to the diversity of maize varieties and the complexity of population growth, future research will enrich the samples and add relevant information to improve the model recognition ability and real-time detection performance. It is planned to port the model to the edge computing platform to optimize its lightweight and high efficiency, so as to achieve accurate and efficient maize growth detection and provide powerful technical support for agricultural production.

## 5. Conclusions

In this study, for lightweight maize growth target detection, based on YOLOv8, the backbone network in the model was replaced with the StarNet network and a lightweight shared convolutional detection head was introduced to reduce the number of parameters in the model and the model size. The CAFM attention module was added to enhance the feature expression capability by capturing long-range dependency and local feature extraction, to solve the problem that the model may have insufficient feature expression capability when dealing with multi-scale targets and complex backgrounds. The C2f module of the neck network was replaced and the C2f module was innovated by using the StarNet network, which reduces the number of model parameters and computation amount while maintaining a high feature extraction capability. The following conclusions can be drawn from the comparative analysis of the experimental results:(1)In this study, we use a lightweight StarNet network and lightweight shared convolutional detection head to reduce the model parameters and computation, and the improved model has a parameter count of 1.8 M and a model size of 3.8 MB, which are suitable for edge devices with limited resources. The model occupies the smallest memory compared to the mainstream target detection models Faster R-CNN, SSD, YOLOv5s, and YOLOv9. LCS-YOLOv8 achieves adaptive detection under different lighting conditions with high model robustness.(2)In order to verify the detection effect of the modified LCS-YOLOv8 network model, this study conducted a visual comparative analysis by setting up three sets of comparison tests. The experimental results show that the improved lightweight LCS-YOLOv8 network model exhibits high detection accuracy and reduces the occurrence of missed detections under the measured light conditions. In addition, by detecting maize leaves and counting the number of leaves, this study helps to improve the automation of leaf number detection work and provides technical support for high-quality assessment of maize growth. Meanwhile, the lightweight model can provide a reference for the deployment of mobile terminal detection equipment to apply intelligent detection of maize growth.

## Figures and Tables

**Figure 1 sensors-24-05279-f001:**
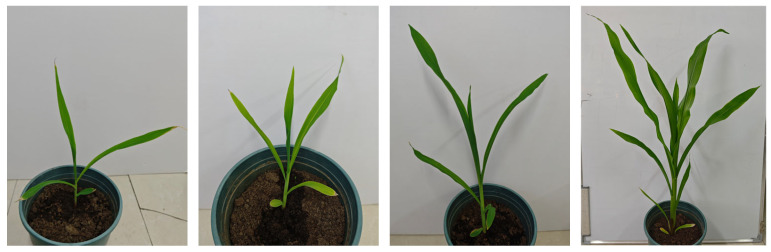
Partial data set (different angles at different times).

**Figure 2 sensors-24-05279-f002:**
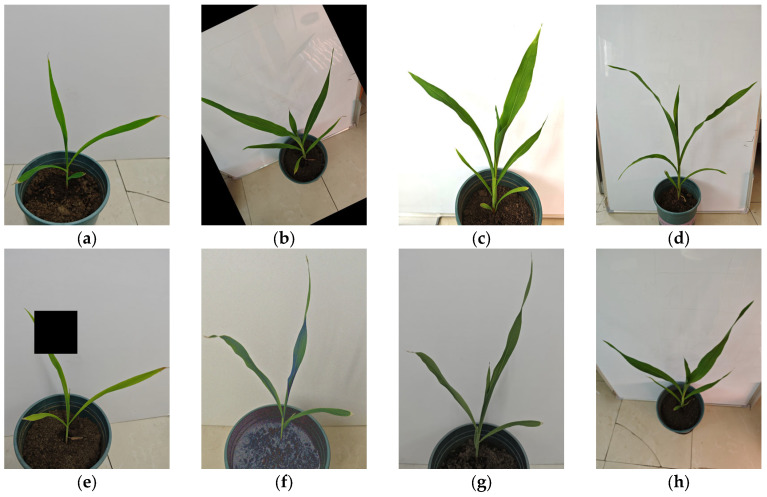
Data enhancement. (**a**) original images; (**b**) random rotation; (**c**) random brightness; (**d**) saturation adjustment; (**e**) random masking; (**f**) noise; (**g**) light changes; (**h**) random blur.

**Figure 3 sensors-24-05279-f003:**
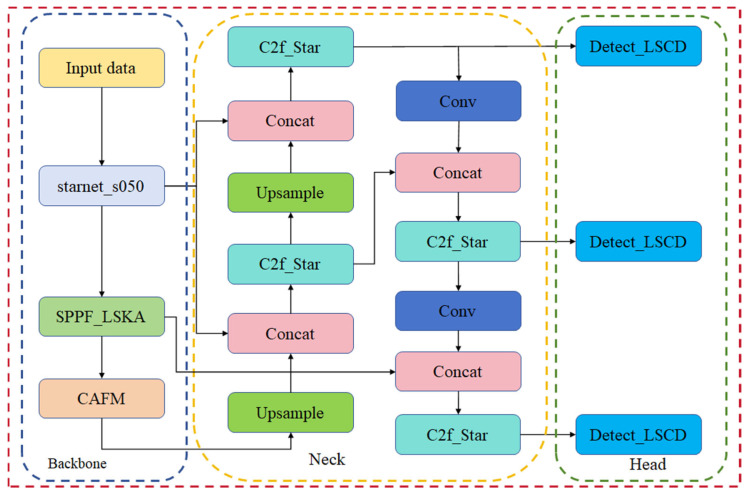
LCS-YOLOv8 model architecture. starnet_s050 replaces the YOLOv8 backbone network. LSKA improves the SPPF module. Adding the CAFM attention module to the backbone network. Improvement of C2f using modules in the StarNet network. Improvement of the detection head with the LSCD module.

**Figure 4 sensors-24-05279-f004:**
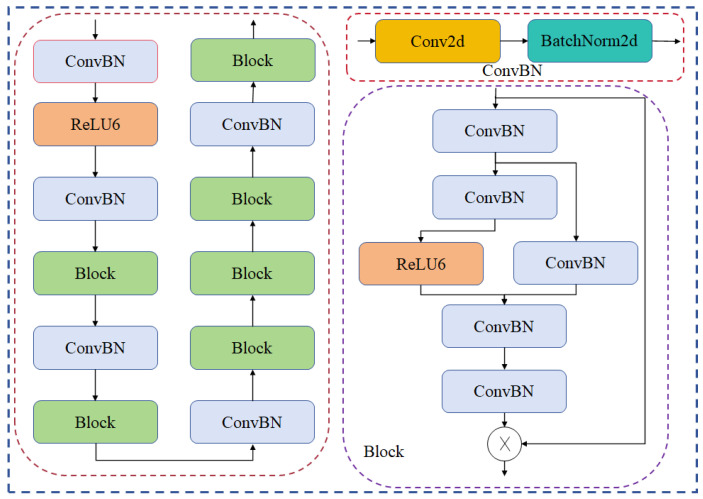
StarNet structure.

**Figure 5 sensors-24-05279-f005:**
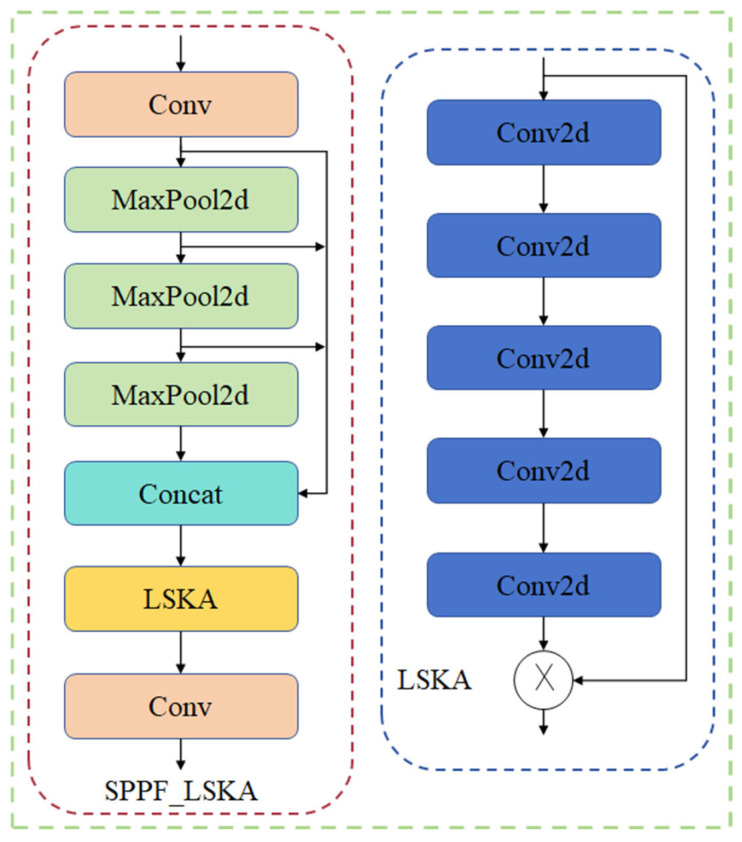
SPPF_LSKA structure.

**Figure 6 sensors-24-05279-f006:**
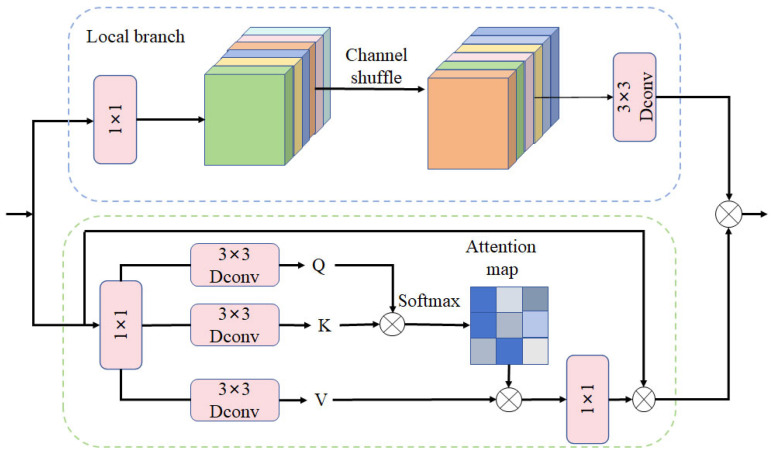
CAFM structure.

**Figure 7 sensors-24-05279-f007:**
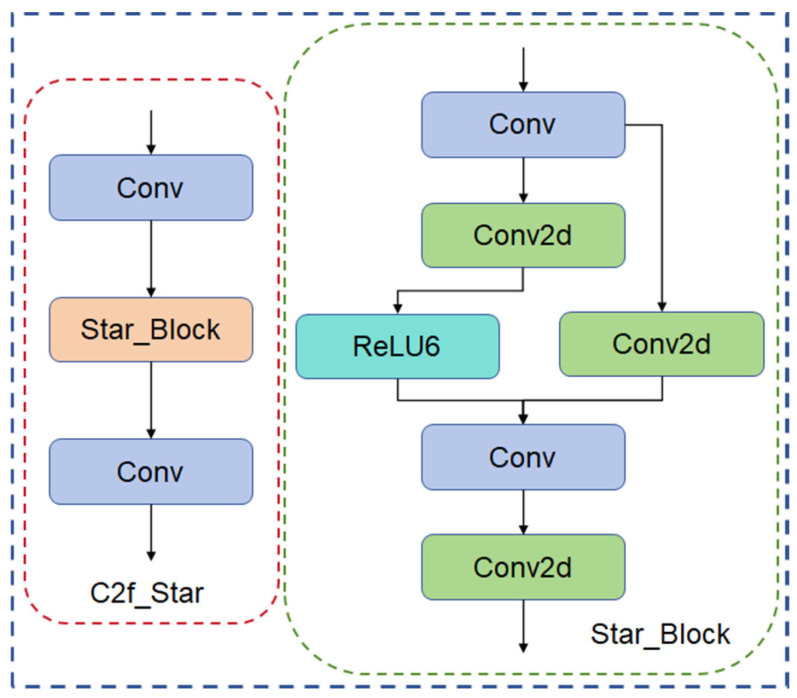
C2_Star structure.

**Figure 8 sensors-24-05279-f008:**
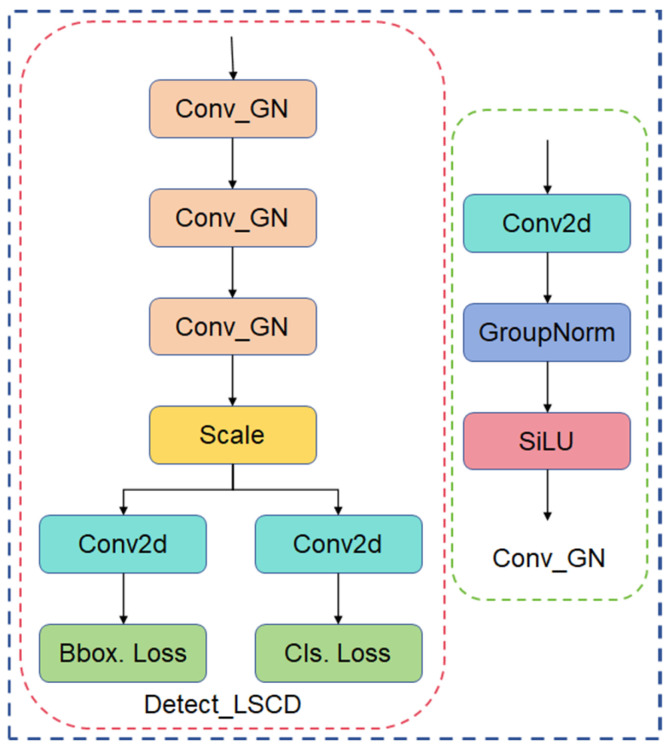
Detect_LSCD structure.

**Figure 9 sensors-24-05279-f009:**
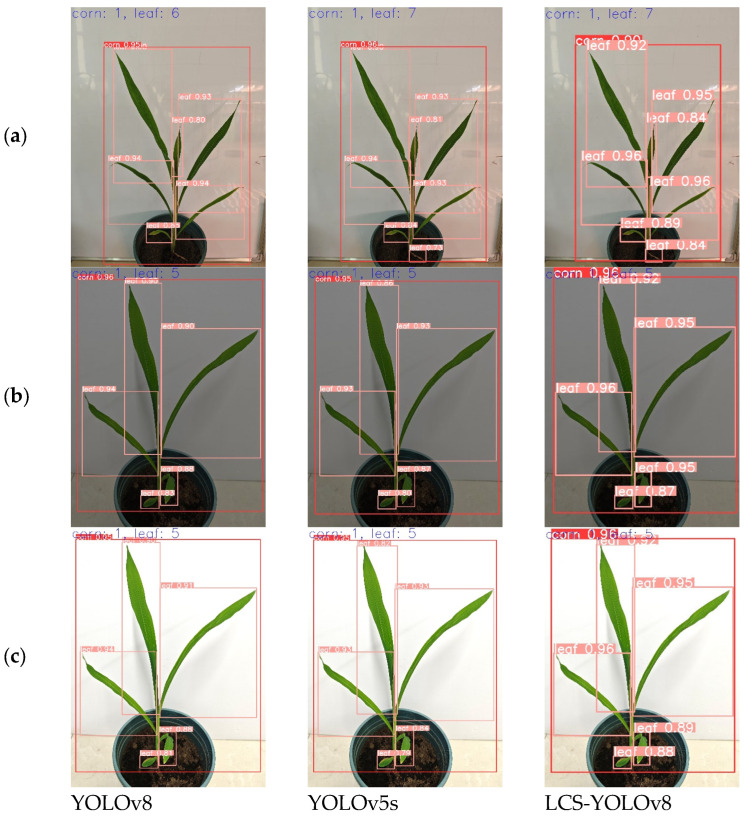
Results of different models for identification of maize growth. (**a**) Image of maize in normal weather. (**b**) Image of corn in cloudy weather. (**c**) Image of corn in sunny weather. The first column shows the results of YOLOv8 in different weather conditions. The second column shows the test results of YOLOv5s in different weather. The third column shows the detection results of LCS-YOLOv8 in different weather conditions.

**Figure 10 sensors-24-05279-f010:**
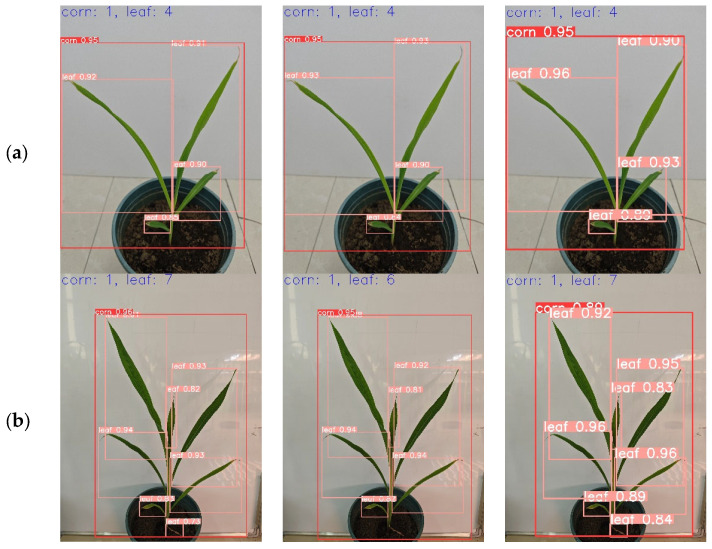
Effect of maize leaf identification at different growth stages. (**a**) Image of maize at 4-leaf stage. (**b**) Image of maize at 7-leaf stage. (**c**) Image of maize at 9-leaf stage. The first column shows YOLOv8’s test results over time. The second column shows the results of YOLOv5s tests over time. The third column shows the test results of LCS-YOLOv8 over time.

**Figure 11 sensors-24-05279-f011:**
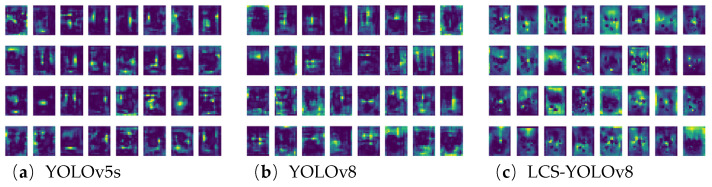
Visualization of different model feature mappings. (**a**) Stages23_C3_features network feature map; (**b**) stages21_C2f_features network feature map; (**c**) stages18_C2f_features network feature map.

**Table 1 sensors-24-05279-t001:** Training parameter settings.

Parameter	Value
epochs	200
patience	50
batch	16
optimizer	SGD
weight_decay	0.0005
momentum	0.937
warmup_momentum	0.8
close_mosaic	10
iou	0.7
imgsz	640
workers	8
lrf	0.01

**Table 2 sensors-24-05279-t002:** Comparison of experimental results.

Model	Precision/%	Recall/%	F1	mAP50/%	FLOPs/G	Params (M)	Size (MB)	FPS
YOLOv5s	98.3	95.9	97.1	98.1	15.8	7.0	14.4	118.2
YOLOv8n	97.8	95.2	96.5	97.8	8.1	3.0	6.3	178.7
YOLOv9	94.9	91.2	92.7	94.3	11.0	2.7	5.8	141.9
Faster R-CNN	76.6	93.9	84.4	92.9	941.0	28.3	108.2	85.6
SSD	98.2	82.0	89.4	93.1	60.9	23.8	91.1	103.4
LCS-YOLOv8	97.9	95.5	96.7	97.5	4.9	1.8	3.8	185.3

**Table 3 sensors-24-05279-t003:** Results of the ablation experiment.

StarNet	LSKA	CAFM	LSCD	C2f-Star	Precision/%	Recall/%	F1	mAP50/%	FLOPs/G	Params (M)	Size (MB)	FPS
					97.8	95.2	96.5	97.8	8.1	3.0	6.3	178.7
√					97.8	95.0	96.4	97.5	6.5	2.2	4.7	180.2
√			√		97.3	95.7	96.5	97.4	5.7	2.1	4.3	175.5
√	√		√		97.9	95.2	96.5	97.7	5.2	1.9	3.9	179.3
√	√			√	98.0	95.2	96.6	97.6	5.4	2.3	4.8	181.4
√		√	√	√	97.7	95.4	96.5	97.4	5.0	2.0	4.1	182.6
√	√	√	√	√	97.9	95.5	96.7	97.5	4.9	1.8	3.8	185.3

## Data Availability

The data presented in this study are available on request from the corresponding authors.

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
