# Peer review of "Lightweight Corn Leaf Detection and Counting Using Improved YOLOv8"

_sensors, 2024, doi:10.3390/s24165279_

Round 1

Reviewer 1 Report

Comments and Suggestions for Authors

This study presents a significant advancement in maize leaf detection by proposing an improved lightweight YOLOv8 method. The integration of the StarNet network and CAFM, along with the enhanced StarBlock module, markedly improves feature representation and detection accuracy.

comment 1: In the introduction section, it is necessary to add literature references regarding the detection algorithms for corn leaves. For example, how did the authors choose between YOLOv5 and YOLOv8? The following references are recommended:

  1. Li, Y., Liao, J., Wang, J., Luo, Y., and Lan, Y., 2023. Prototype Network for Predicting Occluded Picking Position Based on Lychee Phenotypic Features. Agronomy, 13(9), p.2435.

comment 2: In line 146 of page 3, how did the authors ensure the consistency and reliability of the data during the multi-angle acquisition process of maize from seedling to tassel stage, given the potential variations in lighting conditions and other environmental factors?

comment 3: In Figure 2, when the authors performed data enhancement, how did they design the light angles, and how did they determine which augmentations would affect the training data?

comment 4: In Table 2, more details about the LCS-YOLOv8 should be added. Since the framework combines multiple network modules, how can we be sure it will work effectively when these modules are combined?

Comments on the Quality of English Language

Moderate editing of English language required

Reviewer 2 Report

Comments and Suggestions for Authors

The entire text of the article contains incorrect word wrapping, which must be corrected.

The authors did not provide information on which method of growing corn was used in the experiment. Repetition and repeatability of the experiment.

There are no clear characteristics of the mobile phone camera. Does it meet all the requirements for conducting the experiment?

It is not entirely clear why reference should be made to changes in image quality if the photo is taken in a laboratory with artificial lighting. The photo clearly shows camera glare, which could affect the quality of the processed information.

Why were exactly 80% of the photos selected for constructing the training set?

The author refers to photo improvement methods (lines 169-176) but does not indicate which ones were used.

Section 2 does not contain clear literary references to research methods. Formulas do not have references to authorship.

Section 3.2 presents elements of the research methodology that should be given in Section 2.

Tables and figures in the article must be formatted according to the placement rules (figures and tables are placed with a left shift).

In conclusion #2, the authors indicate that the models operate in various weather conditions, which is not true. It is also stated that the models can serve as a standard, which is also not true.

In section 4, there are no references to literary sources, which should be in the discussion.

In the list of references, many sources do not indicate pages.

Round 2

Reviewer 2 Report

Comments and Suggestions for Authors

The authors of the article corrected all remarks. The article can be published. References correspond to literary sources. The conclusions correspond to the obtained results.